# Improvement of SAM2 Algorithm Based on Kalman Filtering for Long-Term Video Object Segmentation

**DOI:** 10.3390/s25134199

**Published:** 2025-07-05

**Authors:** Jun Yin, Fei Wu, Hao Su, Peng Huang, Yuetong Qixuan

**Affiliations:** 1School of Computer Science and Technology, Zhejiang University, Hangzhou 310058, China; 12221242@zju.edu.cn (J.Y.); wufei@zju.edu.cn (F.W.); 2Zhejiang Dahua Technology Co., Ltd., Hangzhou 310053, China; huang_peng@dahuatech.com (P.H.); qi_xuanyuetong@dahuatech.com (Y.Q.)

**Keywords:** SAM 2, long-term video, SA-V, LVOS, Kalman filter

## Abstract

The Segment Anything Model 2 (SAM2) has achieved state-of-the-art performance in pixel-level object segmentation for both static and dynamic visual content. Its streaming memory architecture maintains spatial context across video sequences, yet struggles with long-term tracking due to its static inference framework. SAM 2’s fixed temporal window approach indiscriminately retains historical frames, failing to account for frame quality or dynamic motion patterns. This leads to error propagation and tracking instability in challenging scenarios involving fast-moving objects, partial occlusions, or crowded environments. To overcome these limitations, this paper proposes SAM2Plus, a zero-shot enhancement framework that integrates Kalman filter prediction, dynamic quality thresholds, and adaptive memory management. The Kalman filter models object motion using physical constraints to predict trajectories and dynamically refine segmentation states, mitigating positional drift during occlusions or velocity changes. Dynamic thresholds, combined with multi-criteria evaluation metrics (e.g., motion coherence, appearance consistency), prioritize high-quality frames while adaptively balancing confidence scores and temporal smoothness. This reduces ambiguities among similar objects in complex scenes. SAM2Plus further employs an optimized memory system that prunes outdated or low-confidence entries and retains temporally coherent context, ensuring constant computational resources even for infinitely long videos. Extensive experiments on two video object segmentation (VOS) benchmarks demonstrate SAM2Plus’s superiority over SAM 2. It achieves an average improvement of 1.0 in J&F metrics across all 24 direct comparisons, with gains exceeding 2.3 points on SA-V and LVOS datasets for long-term tracking. The method delivers real-time performance and strong generalization without fine-tuning or additional parameters, effectively addressing occlusion recovery and viewpoint changes. By unifying motion-aware physics-based prediction with spatial segmentation, SAM2Plus bridges the gap between static and dynamic reasoning, offering a scalable solution for real-world applications such as autonomous driving and surveillance systems.

## 1. Introduction

Recent advancements in computer vision have been revolutionized by segmentation models capable of precise pixel-level mask generation. The Segment Anything Model (SAM) [1] has emerged as a foundational paradigm for visual segmentation tasks due to its exceptional ability to produce accurate instance masks. Building upon this framework, SAM2 [2] introduces a streaming memory architecture to sequentially process video frames, thereby maintaining spatial context over long sequences and significantly improving performance in video object segmentation (VOS) [2,3].

Despite these advancements, SAM2 exhibits systematic limitations in visual object tracking (VOT) scenarios, which severely restrict its utility in real-world dynamic environments. These limitations primarily arise from the model’s inadequate capacity to model temporal dynamics, leading to inconsistencies in tracking over prolonged sequences. For instance, SAM2’s reliance on spatial features causes positional drift and mask fragmentation when dealing with rapidly moving or self-occluded objects, as it fails to account for motion cues like velocity or trajectory changes. Similarly, its inability to explicitly model inter-object relationships—such as occlusion, proximity, or separation—results in confusion among visually similar targets in crowded scenes, while the linear decline in multi-object segmentation efficiency further hinders scalability. The streaming memory architecture, while preserving some historical context, lacks adaptive update mechanisms to handle long-term temporal dependencies, leading to error accumulation and eventual target loss in scenarios with drastic viewpoint changes or prolonged sequences [4,5]. Additionally, the model’s poor sensitivity to fine-grained dynamic features, such as edge deformation or texture evolution during occlusion recovery, exacerbates segmentation inaccuracies. Although external cues like human annotations or supplementary trackers can partially alleviate these issues, they demand continuous manual intervention and fail to ensure temporal smoothness, particularly in sequences with abrupt motion or inconsistent mask quality [3]. The root cause of these challenges lies in the fundamental mismatch between SAM2’s static segmentation framework and the dynamic requirements of long-term tracking, multi-target interactions, and evolving feature details in realistic video settings.

Kalman filtering (KF) [6] has long served as a foundational technique in VOT, and is widely adopted in multi-object tracking (MOT) systems [7,8,9,10] due to its efficiency and interpretability. It excels in single-object tracking by modeling motion dynamics and reducing uncertainty through recursive state estimation. In MOT frameworks, KF is often combined with association methods such as JPDA [11] or MHT [12], and more recently with Labeled Random Finite Sets (LRFS) [13], to enhance robustness against occlusion and improve re-identification performance. However, these advanced MOT techniques typically involve complex association logic or require extensive offline optimization, which is not conducive to real-time applications or lightweight deployment. In contrast, we choose the linear Kalman filter as our baseline tracker, not only for its strong performance in motion prediction but also for its compatibility with streaming segmentation frameworks like SAM2 [2].

To address these limitations, we propose SAM2Plus, a training-free enhancement framework designed to improve SAM2’s tracking performance in VOT scenarios. SAM2Plus systematically integrates motion-aware prediction, adaptive evaluation, and scalable memory management to reconcile the static–dynamic inference inconsistency. The framework incorporates a Kalman filter prediction and adaptive update mechanism to anticipate object trajectories and dynamically adjust model states using physical motion constraints, thereby enhancing resilience to partial occlusions or sudden velocity changes. A dynamic thresholds and multi-criteria evaluation system balances model confidence scores with temporal tracking metrics (e.g., motion coherence, appearance consistency), enabling intelligent frame selection that prioritizes robust tracking information while adapting criteria to evolving environmental conditions. Furthermore, an adaptive streaming memory management module employs a novel dynamic library update strategy to selectively retain high-quality historical context while discarding redundant entries, ensuring long-term tracking stability and preventing resource exhaustion, even in sequences with extreme temporal variability.

Experimental validation on two benchmark datasets—SA-V and LVOS—demonstrates the efficacy of SAM2Plus. Quantitatively, the proposed framework achieves an average improvement of 1.0 in J & F metrics across all 24 direct comparisons, with the highest enhancement of 2.3 in J & F scores observed in long-term sequences, as shown in Figure 1. These gains are realized without additional training or external parameters, as the Kalman filter introduces physical dynamics constraints to mitigate positional errors, dynamic thresholds reduce ambiguities in crowded scenes, and memory management ensures computational scalability. By unifying motion-aware prediction with SAM2’s spatial segmentation strengths, SAM2Plus addresses the core limitations of its predecessor, providing a versatile solution for real-world applications such as surveillance, autonomous driving, and sports analytics.

The contributions of this work are threefold: first, a lightweight framework that enhances SAM2’s tracking capabilities without compromising its original performance; second, a systematic integration of motion prediction, adaptive evaluation, and memory management to bridge the gap between static segmentation and dynamic tracking; and third, empirical evidence of superior robustness in challenging scenarios, establishing a novel paradigm for long-term video object segmentation.

The remainder of this paper is organized as follows: Section 2 reviews the related work on VOS and SAM. Section 3 describes the proposed SAM2Plus method, including the integration of Kalman filtering, dynamic thresholds, and multi-metric evaluation, as well as memory management and optimization for long video sequences. Section 4 details the datasets, evaluation metrics, and results, demonstrating the effectiveness of SAM2Plus. Finally, Section 5 discusses the impact of the sub-modules, and Section 6 concludes the paper and outlines potential future work.

## 2. Related Work

### 2.1. Visual Object Segmentation (VOS)

Video object segmentation (VOS) is a core computer vision task that performs pixel-level segmentation and tracking of target objects across video frames, requiring both spatial accuracy in individual frames and temporal consistency throughout sequences [3,14]. Existing approaches can be categorized by their supervision paradigms: Semi-supervised VOS methods, which dominate current research, utilize first-frame annotations as reference. The pioneering STM [15] introduced memory mechanisms for cross-frame matching, while CFBI [16] enhanced robustness through foreground–background feature contrast. Unsupervised approaches like MotionTrack [17] eliminate annotation requirements by leveraging motion cues, although with reduced precision. Traditional graph-based methods showed limitations in complex scenarios, leading to modern deep learning solutions spanning propagation-based OSVOS [18], detection-integrated TrackR-CNN [19], and transformer architectures SAM2 [2]. Current challenges include handling occlusions, appearance variations, and computational efficiency, with recent advances focusing on lightweight designs [3,20] and multimodal fusion [21]. Standard benchmarks (DAVIS [22], YouTube-VOS [23] and MOSE [24], SA-V [2]) and metrics (Jaccard, F-measure) drive progress in this evolving field [2,3]. This paper investigates the application of Segment Anything Model 2 (SAM 2) for semi-supervised VOS.

### 2.2. Segment Anything Model (SAM)

The Segment Anything Model (SAM) [1] is the first foundation model for general image segmentation tasks, aiming to achieve zero-shot open-world segmentation through prompt interaction. Some researchers have made lightweight designs of the original SAM in order to run SAM on resource-constrained embedded devices, thereby reducing the computing cost. FastSAM [25] accelerates the segmentation of all objects in the image by replacing the original transformer architecture with the CNN-based architecture YOLOv8-seg [26,27]. MobileSAM [28] introduces a decoupled distillation method to create a lightweight image encoder derived from SAM. MobileSAMv2 [29] significantly improves the efficiency of SAM by replacing grid search with object-aware prompt sampling. EfficientSAM [30] leverages the SAMI pretraining method to significantly reduce computational costs while maintaining performance comparable to large SAM models, making it more suitable for real-world applications. SAM-Lightening [31] accelerates SAM’s vision backbone through dilated flash attention reconstruction coupled with adaptive inter-layer distillation, specifically targeting latency bottlenecks and memory-intensive computations in dense prediction systems. EfficientViT-SAM employs an EfficientViT-based image encoder to replace SAM’s original heavy encoder, significantly improving segmentation efficiency while maintaining accuracy [32]. EdgeSAM [33] accelerates SAM for edge deployment by distilling its ViT-based image encoder into a CNN architecture, jointly optimizing the prompt encoder and mask decoder to preserve task-aware dynamics.

SAM2 [2] enhances Meta’s original SAM [1] into a unified promptable architecture for both image and video segmentation, introducing a memory bank mechanism to retain object-specific features across frames for consistent single-object tracking, while preserving real-time processing and zero-shot generalization capabilities. Building on this foundation, Ma et al. adapted SAM2 to the medical domain by proposing MedSAM [34], a universal segmentation foundation model trained on 1.57 M multi-modal medical images. Chen et al. proposed SAM2-Adapter [35], the first adapter for Segment Anything 2 (SAM2) that achieves state-of-the-art performance in medical image segmentation, camouflaged object detection, and shadow detection tasks. While SAM2Long [4] enhances long video segmentation via tree-based memory, its computational overhead grows exponentially with higher FPS inputs and deeper memory architectures. SAMURAI [5] enhances tracking performance by introducing a memory cleaning module that dynamically discards irrelevant information. Videnović et al. proposed SAM2.1++ [36], a distractor-aware introspective memory model that achieves significant performance gains in video object segmentation (VOS), outperforming existing methods on six of seven key benchmarks while introducing the DiDi dataset to quantitatively assess tracking stability. Jiang et al. proposed SAM2MOT [37], a Tracking-by-Segmentation paradigm for multi-object tracking that directly generates tracks from SAM2 masks, achieving state-of-the-art performance on DanceTrack (+2.1 HOTA) through novel trajectory management and cross-object interaction modules, while maintaining zero-shot generalization capability. We propose an enhanced framework based on SAM2 to further improve object segmentation capabilities in long videos.

## 3. Methods

As illustrated in Figure 2, we introduce SAM2Plus, an enhanced video segmentation framework built upon Segment Anything Model 2 (SAM2) [2]. The proposed architecture integrates three key improvements: the Kalman Filtering-Intersection Over Union (KF-IoU) fusion framework for temporal stability, the Adaptive Historical Frame Selection Strategy Based on Dynamic Threshold (AHFSS-DT) for efficient memory bank updates, and memory optimization techniques for long-term segmentation scalability.

### 3.1. Baseline SAM2 Overview

We briefly review the SAM2 architecture to establish the foundation for our improvements. SAM2 [2] is a unified framework for interactive image and video segmentation, featuring four core components:Image Encoder: SAM2 employs a ViT-Hiera backbone [38] to extract hierarchical visual embeddings. These dense features are used for both image and video inference.Prompt Encoder: User-defined prompts (e.g., clicks, boxes, and masks) are encoded into task-specific embeddings to guide segmentation.Memory Bank: SAM2 maintains a memory bank of previous frame features to support video tracking. Features are encoded using a memory encoder and aligned through memory attention.Mask Decoder: Based on fused visual and memory features, the mask decoder generates segmentation masks conditioned on prompts. In video sequences, it also ensures temporal stability.

However, the fixed-window memory mechanism and the reliance on IoU-based mask selection in SAM2 result in two limitations: (1) reduced tracking accuracy in occlusion or fast-motion scenarios; and (2) sensitivity to single-frame segmentation errors that can propagate over time.

### 3.2. Kalman Filtering-IoU Fusion Framework

To improve tracking consistency, we propose the Kalman Filtering-IoU (KF-IoU) fusion framework, which introduces a predictive component to stabilize tracking and reduce the impact of single-frame errors. The KF-IoU fusion framework combines the KF for motion estimation with the IoU prediction head of the mask decoder to dynamically refine the object’s tracking state. Specifically, the KF module predicts the object’s bounding box in the current frame based on prior motion dynamics, while the IoU head evaluates the confidence of the predicted mask. These two outputs are then fused through a weighted likelihood function, resulting in a more reliable mask proposal.

The Kalman filter [6] is an optimal recursive estimator for linear dynamical systems in discrete time, iteratively refining state predictions by leveraging posteriori state estimate $\hat{x}$ and current measurements $z_{t}$ while maintaining uncertainty through covariance matrices P. In object tracking applications, its process integrates state transition models F, Kalman gains $K_{t}$, observation models H, process noise Q, and observation noise R terms to alternately predict and update target states at each time step [7,8,10]:(1)$$\begin{array}{c} predict\left\{\begin{array}{c} {\hat{x}}_{t\left|t-1\right.}=F_{t}{\hat{x}}_{t-1\left|t-1\right.}\\ P_{t\left|t-1\right.}=F_{t}P_{t-1\left|t-1\right.}F_{t}^{Τ}+Q_{t} \end{array}\right.\\ update\left\{\begin{array}{c} {K_{t}=P_{t\left|t-1\right.}H_{t}^{Τ}(H_{t}P_{t\left|t-1\right.}H_{t}^{Τ}+R_{t})}^{-1}\\ {\hat{x}}_{t\left|t\right.}={\hat{x}}_{t\left|t-1\right.}+K_{t}\left(z_{t}-H_{t}{\hat{x}}_{t\left|t\right.-1}\right)\\ P_{t\left|t\right.}=\left(I-K_{t}H_{t}\right)P_{t\left|t-1\right.} \end{array}\right. \end{array}$$

The Kalman filter operates in two alternating phases: prediction, which estimates the next state using prior information from the current posterior, and update, which adjusts this estimate based on a new measurement derived via the observation model H.

We employ a standard Kalman filter with constant velocity motion and linear observation model, where the bounding box coordinates $\left(x,y,a,h\right)$ serve as direct observations of the object state. The state vector x is defined as:(2)$$x={\left[x,y,a,h,v_{x},v_{y},v_{a},v_{h}\right]}^{T}$$
where x,y represents the center coordinate of the bounding box, a=w/h represents the aspect ratio of the target, h represents the height of the target, and the initial value is the position of the first frame of the target. $v_{x},v_{y},v_{a},v_{h}$ indicates the change rate of the corresponding variable with time and the initial value is 0.

For each mask $M_{i}$ and its corresponding bounding box, $B_{i}=\left(x_{\min},y_{\min},x_{\max},y_{\max}\right)$ can be calculated by the following mathematical Formula (3):(3)$$\left\{\begin{array}{c} x_{\min}=\underset{\left(i,j\right)\in S}{\min}j\\ x_{\max}=\underset{\left(i,j\right)\in S}{\max}j\\ y_{\min}=\underset{\left(i,j\right)\in S}{\min}i\\ y_{\max}=\underset{\left(i,j\right)\in S}{\max}i \end{array}\right.$$
where $S=\left\{\left(i,j\right)\left|M_{i,j}>0\right.\right\}$ represents the coordinate set of all non-zero (foreground) pixels in the mask.(4)$$\begin{array}{l} iou_{weighted}=\omega \cdot iou_{kf}+\left(1-\omega \right)\cdot iou\\ iou\_inds_{best}=\arg\max\left(iou_{weighted}\right) \end{array}$$
where iou represents the intersection over union (IoU) of the original mask, $iou_{kf}$ represents the IoU between the bounding box predicted by the Kalman filter and the actual detected bounding box (obtained by converting masks through Formula (3)). By calculating $iou_{kf}$, we can evaluate the accuracy of the Kalman filter prediction and combine it with the actual segmentation results to improve the stability and accuracy of tracking.

In multi-frame object tracking, the initialization and update process of the Kalman filter is as follows: First, when the mean and covariance of the Kalman filter are not initialized, or the number of stable frames is zero, the mask with the highest IoU is selected as the initial mask. The bounding box of this mask is then calculated to initialize the Kalman filter. Next, if the number of stable frames is less than the threshold, the boundary of the current frame is predicted first. The mask with the highest IoU is then selected, as choosing the best IoU mask ensures the accuracy of the predicted state. If the IoU of the selected mask is higher than the stability threshold, the mean and covariance matrix of the Kalman filter are updated, and the number of stable frames is incremented. Finally, when the number of stable frames exceeds the threshold and the system enters the stable state update process, the bounding box of the current frame is predicted. The bounding boxes of all candidate masks are calculated, and the mask with the highest weighted IoU (a combination of the IoU from the Kalman filter and the original IoU) is selected to update the mean and covariance matrices of the Kalman filter. The weighted IoU considers the confidence levels of different candidate objects, thereby selecting the best candidate. Before updating the state, it is determined whether the best IoU is below the set threshold. If it is, this indicates that the objects in the current frame do not match the predicted state, and the number of stable frames is reset.

In video sequence tracking, the KF-IoU fusion framework significantly reduces target mask jitter and enhances tracking stability across consecutive frames. By integrating Kalman filter predictions, the framework minimizes the influence of single-frame noise and assists in recovering target positions during partial occlusions. Weighted IoU improves robustness in low SNR environments where the quality of SAM2 model masks is unstable.

The KF-IoU fusion framework extends mask selection from a single frame to spatio-temporal joint optimization, significantly enhancing the stability and robustness of target tracking in consecutive frame scenarios. In contrast, the original SAM2 model selects the mask with the highest IoU without considering temporal continuity or prediction mechanisms, leading to potential instability and susceptibility to single-frame segmentation errors. By introducing the KF to predict the target motion trajectory and combining it with a dynamic weighted IoU decision-making strategy, spatio-temporal consistency in multi-mask output scenarios is significantly improved. Additionally, when the IoU is detected to be below the threshold, the stable frame count is reset to prevent error accumulation, further enhancing the robustness of the tracking system.

### 3.3. Adaptive Historical Frame Selection Strategy Based on Dynamic Threshold

The original SAM2 [2] selects historical frames at fixed time intervals (e.g., one frame per stride), which can introduce noise when frame quality is low and limits its ability to handle long-term occlusions or deformations. In this section, we propose an Adaptive Historical Frame Selection Strategy based on Dynamic Thresholds (AHFSS-DT) to improve the temporal coherence and robustness of frame selection.

Our method dynamically filters frames using a combination of IoU, object confidence scores, and motion prediction scores, ensuring selected frames maintain high semantic and spatial relevance. As shown in Figure 3, the strategy performs reverse traversal of historical frames to prioritize those that meet dynamically adjusted quality thresholds, while applying a fallback policy—including recent frame retention and gradual threshold relaxation—when valid frames are insufficient. This design significantly enhances tracking continuity and segmentation accuracy in complex dynamic environments.

To address the limitations of fixed thresholds in complex scenarios, we designed a dynamic threshold generation function:(5)$$\begin{array}{c} \tau_{IoU}^{\left(t\right)}=\alpha \cdot E[IoU_{hist}^{\left(t\right)}]+(1-\alpha )\cdot \tau_{IoU}^{\mathrm{base}}\\ \tau_{obj}^{\left(t\right)}=\beta \cdot \left(\underset{i<j}{\max}{\mathrm{ObjScore}}_{j}-{\mathrm{ObjScore}}_{i}\right)+\tau_{obj}^{\mathrm{base}}\\ \tau_{KF}^{\left(t\right)}=\gamma \cdot \frac{\varepsilon_{KF}}{\sigma_{\varepsilon }}+\tau_{KF}^{\mathrm{base}} \end{array}$$
where α,β,γ are adaptive weight coefficients, E[⋅] represents the statistical expectation of the *IoU* of historical frames, and $\tau_{base}$ is the preset baseline threshold. This function achieves dynamic adjustment of the IoU threshold $\tau_{IoU}$ and the object score threshold $\tau_{obj}$ by fusing historical statistical information with the feature differences of the current frame. For the motion prediction score $\tau_{KF}$, we further introduce the Kalman filter prediction error $\varepsilon_{KF}$ as a constraint, where $\sigma_{\varepsilon }$ is the standard deviation of the error.

The validity of the historical frame $f_{i}$ is determined by the following conditions simultaneously:(6)$$\left\{\begin{array}{c} IoU\left(f_{t},f_{i}\right)\geq \tau_{IoU}^{\left(t\right)}\\ ObjScore\left(f_{i}\right)\geq \tau_{obj}^{\left(t\right)}\\ KFConsistency\left(f_{i}\right)\geq \tau_{KF}^{\left(t\right)} \end{array}\right.$$

Through this joint constraint, it is ensured that the screened frames meet the dynamic standards in the three dimensions of spatial overlap degree, detection confidence, and motion consistency. When the number of valid frames $N_{valid}<N_{\max}$, the system starts the fallback policy: (1) recent frame forced retention, forcing the most recent frame $f_{t-1}$ to be included in the candidate set; (2) gradient relaxation filling, reverse traversing historical frames and gradually reducing the threshold $\tau \leftarrow \delta \cdot \tau ,\;\delta \in \left(0,1\right)$ until $N_{valid}=N_{\max}$. where $N_{\max}$ is the maximum number of frames to look back. This multi-dimensional quality assessment mechanism enables the model to dynamically filter noise frames and prioritize the retention of historical information with high semantic relevance and motion coherence.

The proposed AHFSS-DT enhances the cross-frame attention mechanism’s ability to utilize temporal context efficiently. Compared to fixed-interval strategies, our method introduces minimal overhead in threshold computation ($O\left(D\right)$, where D is the feature dimension) and conditional checks $O\left(N_{hist}\right)$, resulting in an overall complexity of $O\left(T+N_{hist}\right)$. By constraining the encoder input sequence length $N_{\max}$, we avoid redundant computation while maintaining high performance in long-term tracking and occlusion recovery tasks.

### 3.4. Memory Management and Optimization for Efficient Video Segmentation

In this section, we present a comprehensive set of optimizations to manage memory efficiently during video segmentation tasks using the SAM2. These optimizations aim to maintain constant VRAM and RAM usage, even for infinitely long video sequences, while preserving the accuracy and efficiency of the original model.

#### 3.4.1. Dynamic Memory Management

In processing long video sequences, memory growth can become unbounded if outdated frames and intermediate results are not effectively released. We address this issue through the following mechanisms:Offloading frames to CPU RAM: By setting the offload_video_to_cpu parameter in the init_state method [39], video frames are transferred from GPU VRAM to CPU RAM, reducing the GPU memory footprint without sacrificing availability during inference.Manual VRAM release after memory attention calculation: Intermediate variables generated during the Memory Attention computation were found to persist unnecessarily in VRAM. We added explicit cleanup operations to release these variables immediately after use, significantly lowering peak memory usage.

#### 3.4.2. Efficient Propagation and Result Management

To ensure prompt correction across video frames while minimizing computational overhead, we apply the following propagation and result management techniques:Limiting propagation length: We cap the propagation length at M, ensuring that each correction operation only affects a fixed number of historical frames. This balances correction robustness with computational cost, reducing the total number of frames processed during inference from N to approximately(M/K) * N, where K is the frame buffer size.Immediate release of processed segmentation results: We observed that caching segmentation results in the video_segments dictionary led to linear memory growth. To counteract this, we implemented proactive deallocation, releasing each frame’s segmentation results immediately after use. This prevents unbounded dictionary expansion and stabilizes memory usage.

#### 3.4.3. Continuous Old Frame Clearing

As shown in Figure 4, we introduce a dynamic mechanism to continuously evict outdated frame data during inference on long or infinite video sequences. This is governed by the max_inference_state_frames parameter, which determines the number of most recent frames to retain in memory.

Retention Threshold Mechanism: Only frames beyond the max_inference_state_frames threshold from the current frame are cleared. Crucially, max_inference_state_frames [39,40] must be greater than max_frame_num_to_track in the propagate_in_video() function to ensure that historically relevant frames are retained until no longer needed.Automatic Cleanup After Propagation: Following each propagation step, frames exceeding the retention limit are automatically released from memory. Frames stored in the preload memory bank are preserved indefinitely.Memory Bounds During Inference: The upper bound of VRAM usage is dictated by the memory required for newly processed frames, retained frames, and the fixed-size preload memory bank. The lower bound corresponds to the memory required for the maximum propagation length plus the preload bank. This strategy ensures memory usage remains within a predictable range, effectively balancing efficiency and safety.

#### 3.4.4. Advanced Memory Optimization Techniques

To further reduce linear memory growth and maintain constant VRAM consumption during prolonged inference, we employ the following enhancements:Index Mapping Decoupling: We refactored frame access logic to use an independent frame index mapping table, decoupling logical frame references from physical storage. This allows precise tracking of required frames without relying on sequential indexing.Dynamic State Synchronization: The frame index mapping table is dynamically maintained during state initialization and updates, ensuring real-time alignment between logical indices and the current frame sequence.Coordinated Cleanup Strategy: A dedicated function systematically removes outdated frames that exceed retention thresholds. This function synchronizes updates to both the image cache container and the logical index table, preserving data-reference consistency.Global Frame Statistics: We modified the historical frame counter to track the total number of frames ever loaded rather than just those currently active. This distinction enables accurate memory accounting and avoids overestimating active storage needs.

These steps ensure that VRAM usage remains constant, as outdated frames are immediately released after their relevance expires, while the preload memory bank and propagated frames are retained within bounded limits. The system avoids linear memory growth by dynamically managing frame access and deletion through the index mapping and state synchronization mechanisms.

By implementing these optimizations, we achieve a robust and efficient memory management system for the SAM2 model. This system ensures that the VRAM and RAM usage remain constant, even when processing infinitely long video sequences. The pipeline maintains the same efficiency and accuracy as the original SAM2, while reducing performance overhead through a series of engineering enhancements. These optimizations are particularly useful for applications with limited GPU memory and for processing long video sequences.

## 4. Experiments

### 4.1. Metrics and Datasets

To assess our method, we utilize two widely-adopted VOS benchmarks [2,14] and use evaluation metrics: region similarity J (IoU-based), contour accuracy F (harmonic mean of precision and recall), and their arithmetic mean J&F as an overall performance indicator [3]. Experiments are performed in a semi-supervised framework where the initial frame’s mask is provided. Specifically, region similarity is calculated as the IoU between predicted and ground-truth masks, while contour accuracy evaluates boundary alignment via the harmonic mean of precision and recall. The combined metric averages these values to summarize system efficacy, where $\hat{M},M$ are the predicted segmentation masks and the ground truth masks, respectively.(7)$$J=\frac{\hat{M}\cap M}{\hat{M}\cup M}$$(8)$$F=\frac{2P_{c}R_{c}}{P_{c}+R_{c}}$$(9)$$J\&F=\frac{J+F}{2}$$

We will use the following two VOS datasets to evaluate our method:

SA-V (Segment Anything in Video) [2] is a large-scale video segmentation benchmark specifically designed for promptable visual analysis across diverse scenarios, comprising 50.9 K video clips with 35.5 million high-precision masks and 642.6 K localized masklets (short-term object instances). This dataset emphasizes challenging real-world segmentation tasks, including persistent occlusions, small objects (occupying < 5% of frame area), and long-term reappearance of objects under varying appearances. To ensure comprehensive model evaluation, SA-V is partitioned into 45,495 training videos (89.4% of total data) for robust feature learning, 155 validation videos (293 masklets) for hyperparameter optimization, and 150 testing videos (278 masklets) for rigorous generalization assessment. By prioritizing spatio-temporal consistency and minimizing data leakage, SA-V establishes a critical benchmark for advancing segmentation robustness in dynamic environments, particularly in handling discontinuous object trajectories and cross-frame ambiguities.

The LVOS v1 [22] benchmark pioneers long-term video object segmentation in practical scenarios, comprising 720 video clips (296,401 frames, 407,945 annotations) with an average duration exceeding 60 s. It introduces critical challenges such as long-term object reoccurrence and cross-temporal interference from similar objects, partitioned into 120 training, 50 validation, and 50 testing videos. Building upon this foundation, LVOS v2 [22] extends the dataset to 420 training, 140 validation, and 160 testing videos, fully incorporating all sequences from LVOS v1 while adding new sequences for enhanced complexity. Covering 44 categories of real-world scenarios, LVOS v2 deliberately excludes 12 categories from the training set to rigorously evaluate the generalization capability of VOS models, particularly in handling unseen object types. This version serves as the primary benchmark in our work due to its expanded scale, comprehensive coverage of LVOS v1 data, and emphasis on real-world robustness evaluation.

### 4.2. Environment and Parameters

All experiments are performed on an NVIDIA A40 GPU with 48 GB of memory, using Python 3.10, PyTorch 2.5.1, CUDA 12.4, and Ubuntu 22.04. The Kalman filter is initialized using the bounding box associated with the best segmentation mask, and stable tracking is achieved after 15 consecutive frames. The tracking system uses the following hyperparameters: a stable IoU threshold of 0.3, a minimum object score logit of 0.3, and a Kalman filter confidence weight of 0.25. For the memory bank, the IoU threshold is set to 0.5, and both the object score and Kalman filter score thresholds are set to 0.0. All remaining parameters are inherited from the default settings in SAM2.

### 4.3. Quantitative Results

The quantitative results presented in Table 1, Table 2 and Table 3 provide a comprehensive comparison of the performance of our proposed method (SAM2Plus) against SAM2 and SAM2Long across different datasets and model sizes.

SA-V Datasets (Table 1): The SA-V dataset consists of short videos (approximately 10 s in length). In these scenarios, the performance improvements of our method, SAM2Plus, are less pronounced compared to SAM2 and SAM2Long. For example, in the SA-V Val (Large) set, SAM2.1Plus achieves a score of 79.6 (J&F), which is slightly better than SAM2.1 (78.6) but not as significant as the improvements seen in longer videos. This is because short videos have fewer frames, and the benefits of our adaptive historical frame selection strategy based on dynamic thresholds (AHFSS-DT) are less evident in such scenarios. However, despite the smaller improvement in J&F scores, SAM2Plus maintains a similar memory usage and inference speed compared to SAM2, making it more practical for real-time applications.

LVOS v2 Datasets (Table 2): The LVOS v2 dataset consists of longer videos, and our method, SAM2Plus, demonstrates more significant performance improvements. The adaptive historical frame selection strategy based on dynamic thresholds (AHFSS-DT) and the integration of KF to predict the target’s motion trajectory significantly enhance the efficiency of cross-frame attention mechanisms, leading to better adaptability to dynamic and complex scenes. For instance, in the LVOS v2 Val (Base+) set, SAM2.1Plus achieves a score of 86.3 (J&F), compared to 83.1 for SAM2 and 85.2 for SAM2Long. While SAM2Long shows a higher J&F score in the LVOS v2 Val (Tiny, Small) set, it comes at a significant cost in terms of memory usage and inference speed, as shown in Table 3.

The evaluation of 1080p videos from the LVOS v2 dataset under A40 GPU settings demonstrates that SAM2Plus achieves a well-balanced trade-off between memory usage and inference speed. Compared to the original SAM2, SAM2Plus slightly increases GPU memory consumption (e.g., 5503 MB vs. 5499 MB with the Large backbone), but maintains comparable inference speed (13.46 FPS vs. 13.13 FPS). This indicates that the additional modules introduced in SAM2Plus (e.g., the KF-IoU fusion and enhanced memory attention) have a minimal impact on frame rate, while contributing to improved tracking consistency. In contrast, SAM2Long, which extends the memory window further to model longer temporal contexts, achieves higher segmentation accuracy (J&F scores) in some cases, but at the cost of significantly increased memory usage (up to 6913 MB) and reduced inference speed (8.51 FPS). This trade-off makes it less suitable for real-time or extended-duration video processing. Therefore, SAM2Plus is considered a more practical and balanced solution for real-time applications, provided that sufficient GPU memory is available. It delivers enhanced performance over the baseline without sacrificing frame rate, and is particularly effective in production environments where moderate resource allocation and high tracking accuracy are both prioritized.

State-of-the-Art Comparison (Table 4): Table 4 presents a performance comparison of various methods on the LVOS v2 validation set. From the table, it is evident that SAM2.1Plus performs the best overall, achieving a J&F score of 86.3. It excels in both seen and unseen categories, with J scores of 81.6 and 83.3, and F scores of 89.0 and 91.2, respectively. Although SAM2.1Long scores slightly higher in some categories, particularly in the unseen category, its significantly increased resource consumption makes it less suitable for real-time applications. In contrast, SAM2.1Plus not only approaches the highest performance but also demonstrates excellent resource efficiency, making it the best choice for real-time applications and long video analysis in production environments. Other methods, such as SAM2Plus and SAM2.1, also show good performance but are slightly inferior to SAM2.1Plus. Overall, SAM2.1Plus achieves the best balance between performance and resource efficiency.

These results highlight the robustness and adaptability of our proposed method, SAM2.1Plus, particularly in long video tracking and occlusion recovery tasks. The improvements in performance on the LVOS v2 dataset, which consists of longer videos, underscore the effectiveness of our method in handling dynamic and complex scenes. In contrast, the performance gains on the SA-V dataset, which consists of shorter videos, are less pronounced, but still demonstrate the overall improvement in tracking continuity and segmentation quality. Additionally, the efficient memory usage and inference speed of SAM2.1Plus make it a more practical choice for real-world applications compared to SAM2Long.

### 4.4. Qualitative Results

As illustrated in Figure 5, the visualization results highlight the performance differences between our method, SAM2, and SAM2Long in various scenarios. When the target of interest is occluded or temporarily disappears and then re-enters the region of interest, SAM2 fails to maintain the target’s tracking. Even upon re-emergence, SAM2 is unable to resume tracking. In contrast, our method can immediately resume tracking the target as soon as it reappears after short-term occlusions or disappearances. This capability is attributed to the integration of KF and multi-frame tracking strategies, which enable accurate position prediction and rapid tracking recovery.

In scenarios where the target moves rapidly, SAM2 exhibits lower segmentation quality and is prone to target loss or inaccurate segmentation. SAM2Long shows some improvements but still suffers from suboptimal segmentation quality in such scenarios. Our method, however, achieves higher segmentation quality in rapid movement scenarios. By introducing KF to predict the target’s motion trajectory and combining it with a dynamic weighted IoU decision strategy, our method effectively mitigates the impact of single-frame noise, ensuring consistent and accurate tracking.

In complex dynamic environments, SAM2’s tracking and segmentation performance often deteriorates due to background interference and target occlusions. While SAM2Long offers some improvement, it still struggles in such scenarios. To address these limitations, we propose an AHFSS-DT for video object tracking. This strategy enhances the cross-frame attention mechanism’s ability to exploit temporal context by incorporating dynamic thresholds, multi-metric evaluation, and flexible frame selection. As a result, our method achieves improved adaptability to dynamic scenes and better performance in long video tracking and occlusion recovery. The visualizations in Figure 5 further demonstrate its superior tracking continuity and segmentation quality in challenging scenarios involving rapid motion, occlusion, and target reappearance—highlighting its robustness and practical potential.

### 4.5. Long Video Segmentation in Real Scenes with SAM2Plus

To evaluate the performance of our proposed SAM2Plus in long video segmentation tasks, we conduct all experiments on a real-world 4K-resolution video stream captured from typical urban environments, including parks and roadways. The video has a frame rate of 25 fps and a total duration of approximately 5–10 min. For computational efficiency, we down-sample the video to 8 fps, which strikes a balance between processing speed and temporal resolution.

All experiments are performed on an NVIDIA A40 GPU with 48 GB of memory, using Python 3.10, PyTorch 2.5.1, CUDA 12.4, and Ubuntu 22.04. We utilize the YOLOv11 detection model [47], introduced by Ultralytics [26] in 2024, to generate object bounding boxes in the first frame. YOLOv11 builds upon the architecture of YOLOv8 [27] with significant improvements in both model design and training methodology, making it well-suited for industrial deployment. In our setup, the detected boxes correspond to human subjects, non-motorized vehicles, and motorized vehicles, which are then converted into masks and used as initial prompts for SAM2.1-Base+. Thanks to the robust correction mechanism in SAM2Plus, any newly introduced prompts can be retroactively applied to all previously inferred frames, enabling iterative refinement and reducing mask drift over time. This allows us to achieve effective object tracking and segmentation throughout the video sequence using only a single frame of initial annotations.

From Figure 6, it is evident that, in dense target scenes, the object masks obtained from the detection boxes in the first frame can overlap. However, SAM2Plus can correct these mask deviations within about 5 frames, resulting in accurate and stable tracking of the true object masks. Additionally, the system can effectively re-track targets that experience short-term occlusions or reappear after disappearing. This capability further demonstrates the robustness of our method in handling long-term object segmentation and tracking. For the target segmentation of one video channel, the video memory of the SAM2.1-Base+ model stabilized at 1550 MB, indicating efficient memory usage.

The combination of YOLOv11 for initial object detection and SAM2.1-Base+ for segmentation and tracking proved to be highly effective for low-speed scene analysis. The system achieved high accuracy, efficient processing, and robust error correction, making it suitable for industrial deployment in various applications such as smart parks and traffic management.

## 5. Discussion

We conducted a series of ablation experiments on the LVOS v2 validation set using SAM2-Base+ as the default model size to observe the influence of each module on the overall segmentation results. It can be seen from Table 5 that both of these two modules have had a positive impact on the SAM2 model. The KF-IoU fusion framework increased by approximately 1%, and the AHFSS-DT module increased by 1.7%. The combination of the two on the LVOS v2 dataset can achieve the best J&F, which are 82.9% and 86.3 respectively. The results of the above ablation experiments illustrate that the KF-IoU module significantly reduces the target mask jitter and enhances the tracking stability across consecutive frames. Meanwhile, the AHFSS-DT module significantly improves the efficiency of the cross-frame attention mechanism. The adaptability to dynamic and complex scenes has been improved, and better performance has been achieved in long video tracking and occlusion recovery tasks.

## 6. Conclusions

In this work, we present SAM2Plus, a zero-shot enhancement framework designed to overcome the long-term tracking limitations of Segment Anything Model 2 (SAM 2) in video object segmentation (VOS). By unifying physics-based motion prediction via the Kalman filter, dynamic quality thresholds, and a novel adaptive memory management system, SAM2Plus achieves significant improvements in segmentation accuracy and reliability across extended video sequences. First, the Kalman filter component models object motion dynamics using physical constraints to predict trajectories and dynamically refine segmentation states, effectively mitigating positional drift during occlusions or abrupt velocity changes. Second, prioritizing high-quality frames and balancing confidence scores with temporal smoothness, dynamic thresholds are coupled with multi-criteria metrics (e.g., motion coherence, appearance consistency), which reduce ambiguities in complex scenes such as crowded environments or those involving fast-moving objects. Then, the optimized memory architecture also ensures computational efficiency by dynamically pruning outdated or low-confidence segmentation records while preserving temporally coherent spatial context, thereby maintaining constant memory usage even for infinitely long videos. Finally, extensive experiments on two VOS benchmarks confirm that SAM2Plus significantly outperforms SAM 2, achieving an average 1.0 improvement in J&F metrics across 24 comparisons and exceeding it by over 2.3 points on SA-V and LVOS datasets for long-term VOS, while enabling real-time operation without additional parameters or fine-tuning and enhancing adaptability to dynamic environments. In the future, we will further optimize the target occlusion problem and explore the semantic interaction among multiple targets within the same framework. Overall, SAM2Plus represents a significant advancement in VOS, offering a powerful tool for a wide range of applications in both images and videos, particularly in dynamic and complex environments.

## Figures and Tables

**Figure 1 sensors-25-04199-f001:**
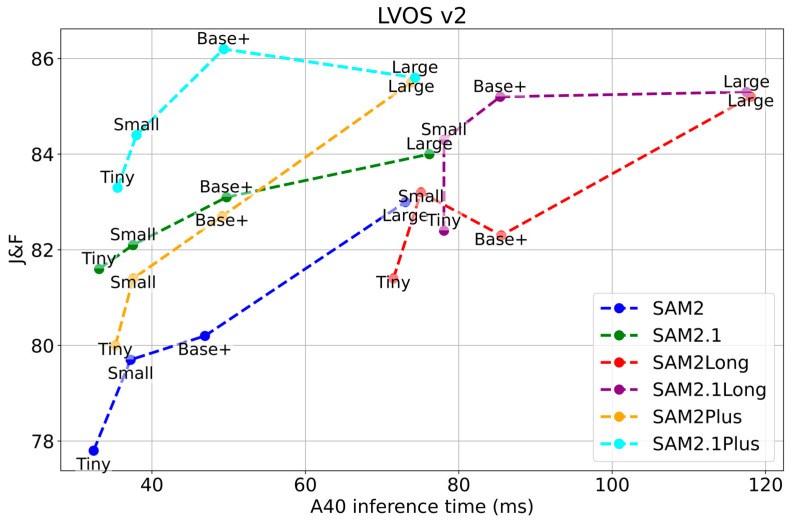
Comparative evaluation of SAM2Plus on the LVOS v2 dataset relative to advanced baselines.

**Figure 2 sensors-25-04199-f002:**
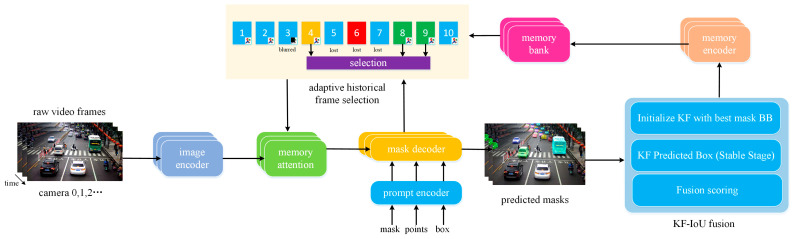
Overview of our SAM 2Plus architecture.

**Figure 3 sensors-25-04199-f003:**
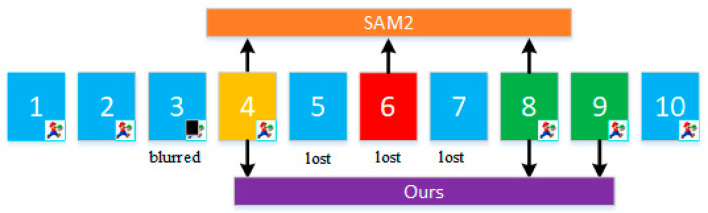
Adaptive historical frame selection. Green indicates highly correlated frames (IoU, object score, and KF score all meet the standards); yellow indicates partially valid frames (IoU meets the standard, but object score or KF score is insufficient); red indicates low-quality frames (IoU or content-independent).

**Figure 4 sensors-25-04199-f004:**
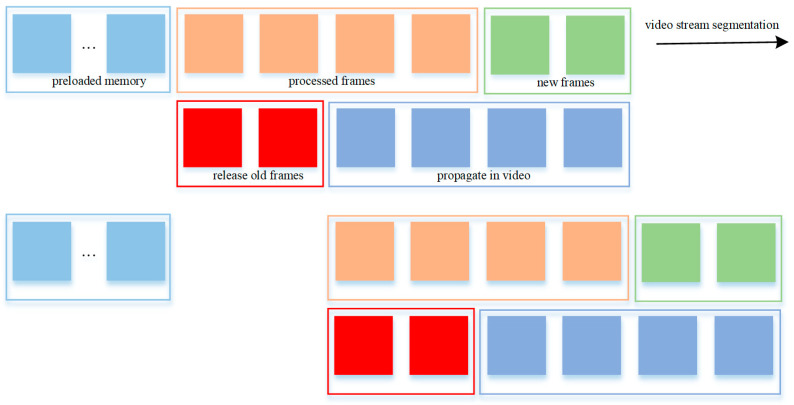
Continuous old frame clearing process.

**Figure 5 sensors-25-04199-f005:**
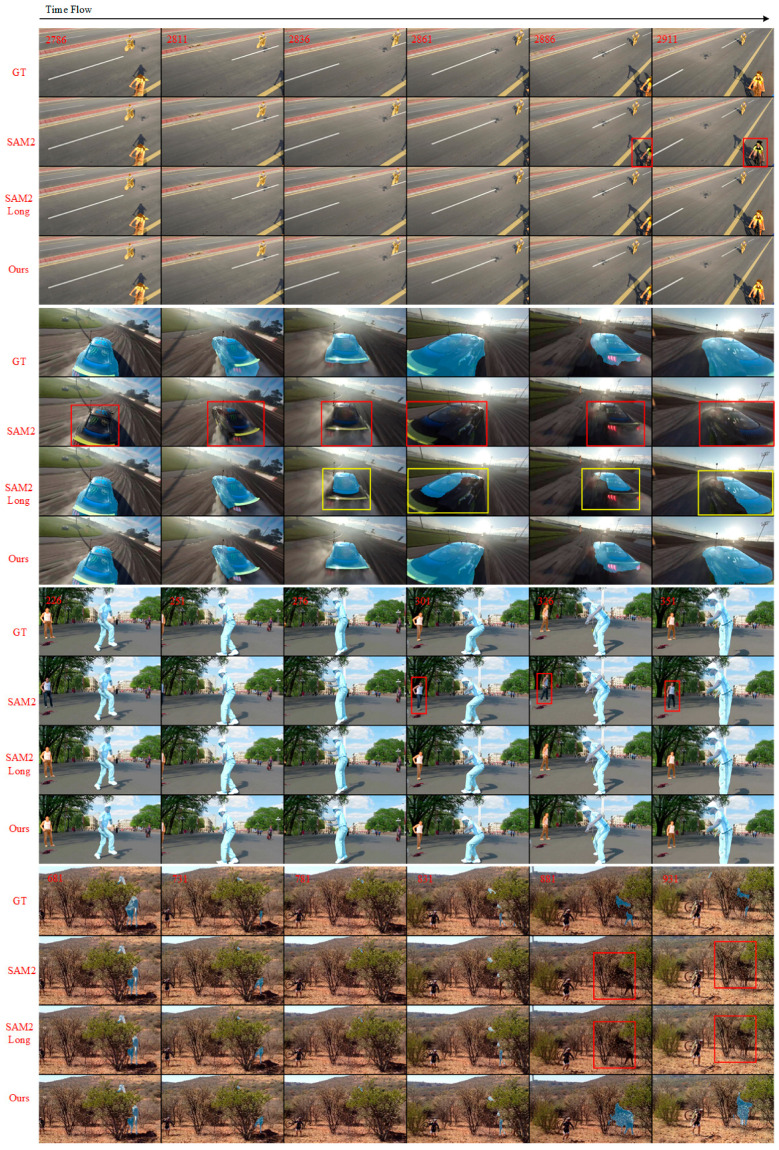
Qualitative comparison of SAM2, SAM2Long and our method, with GT (ground truth) as the reference. The red boxes represent the missing objects and the yellow boxes are used to highlight the low-quality segmentation results.

**Figure 6 sensors-25-04199-f006:**
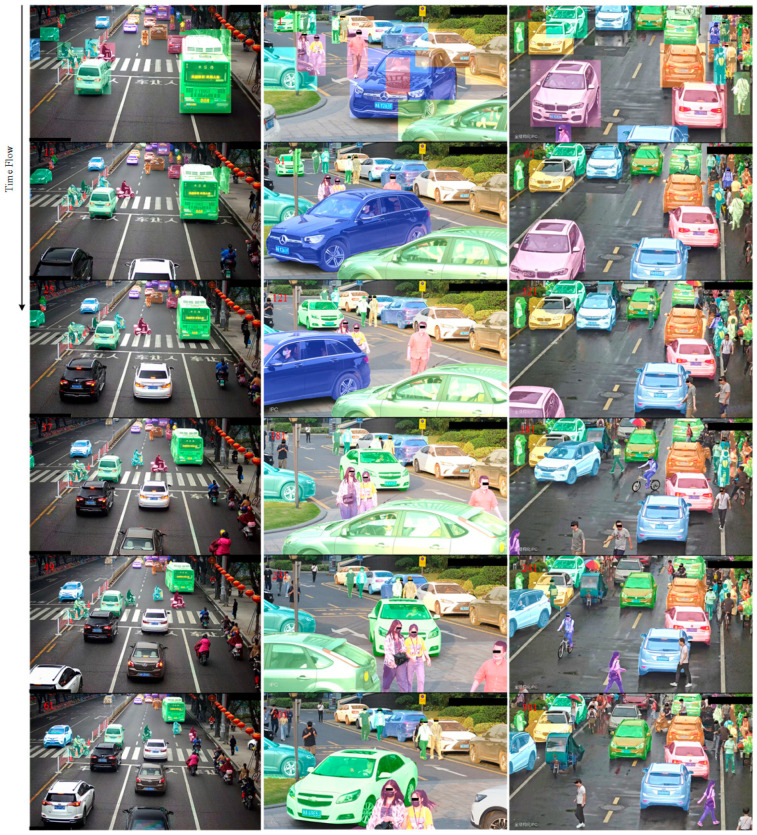
Long video segmentation and tracking results using SAM2Plus based on the detection box in the first frame. The red numbers in the picture represent the video frame numbers.

**Table 1 sensors-25-04199-t001:** Performance comparison of SA-V datasets (SAM2, SAM2Long, and SAM2Plus) across all model sizes.

Method	Backbone	SA-V Val			SA-V Test		
		J&F	J	F	J&F	J	F
SAM2	Tiny	73.5	70.1	76.9	74.6	71.1	78.0
SAM2Long		77.0	73.2	80.7	78.7	74.6	82.7
SAM2Plus		74.3	70.4	78.1	76.4	72.4	80.3
SAM2.1		75.1	71.6	78.6	76.3	72.7	79.8
SAM2.1Long		78.9	75.2	82.7	79.0	75.2	82.9
SAM2.1Plus		75.5	71.7	79.4	76.7	72.8	80.7
SAM2	Small	73.0	69.7	76.3	74.6	71.0	78.1
SAM2Long		77.7	73.9	81.5	78.1	74.1	82.0
SAM2Plus		74.3	70.3	78.2	75.4	71.4	79.4
SAM2.1		76.9	73.5	80.3	76.9	73.3	80.5
SAM2.1Long		79.6	75.9	83.3	80.4	76.6	84.1
SAM2.1Plus		77.3	73.5	81.1	77.6	73.8	81.5
SAM2	Base+	75.4	71.9	78.8	74.6	71.2	78.1
SAM2Long		78.4	74.7	82.1	78.5	74.7	82.2
SAM2Plus		75.4	71.6	79.2	77.1	73.1	81.1
SAM2.1		78.0	74.6	81.5	77.7	74.2	81.2
SAM2.1Long		80.5	76.8	84.2	80.8	77.1	84.5
SAM2.1Plus		77.4	73.6	81.2	78.3	74.5	82.1
SAM2	Large	76.3	73.0	79.5	75.5	72.2	78.9
SAM2Long		80.8	77.1	84.5	80.8	76.8	84.7
SAM2Plus		77.4	73.6	81.2	79.0	74.9	83.0
SAM2.1		78.6	75.1	82.0	79.6	76.1	83.2
SAM2.1Long		81.1	77.5	84.7	81.2	77.6	84.9
SAM2.1Plus		79.6	75.8	83.4	79.6	75.8	83.4

**Table 2 sensors-25-04199-t002:** Performance comparison of LVOS v2 datasets (SAM2, SAM2Long, and SAM2Plus) across all model sizes.

Method	Backbone	LVOS v2 Train			LVOS v2 Val		
		J&F	J	F	J&F	J	F
SAM2	Tiny	94.5	92.9	96.1	77.8	74.5	81.2
SAM2Long		95.2	93.6	96.9	81.4	77.7	85.0
SAM2Plus		95.2	93.6	96.9	80.0	76.3	83.6
SAM2.1		95.0	93.4	96.7	81.6	77.9	85.2
SAM2.1Long		95.2	93.6	96.9	82.4	78.8	85.9
SAM2.1Plus		95.1	93.5	96.8	83.5	79.7	87.3
SAM2	Small	94.6	92.9	96.2	79.7	76.2	83.3
SAM2Long		95.2	93.5	96.9	83.2	79.5	86.8
SAM2Plus		95.1	93.5	96.8	81.4	77.7	85.2
SAM2.1		95.4	93.7	97.0	82.1	78.6	85.6
SAM2.1Long		95.6	94.0	97.3	84.3	80.7	88.0
SAM2.1 Plus		95.4	93.7	97.0	84.8	81.0	88.5
SAM2	Base+	95.3	93.7	96.9	80.2	76.8	83.6
SAM2Long		95.5	93.9	97.2	82.3	78.8	85.9
SAM2 Plus		95.5	93.9	97.2	82.9	79.3	86.5
SAM2.1		95.6	94.0	97.2	83.1	79.6	86.5
SAM2.1Long		95.6	94.0	97.2	85.2	81.5	88.9
SAM2.1Plus		95.6	94.0	97.2	86.3	82.6	90.0
SAM2	Large	95.2	93.6	96.8	83.0	79.6	86.4
SAM2Long		95.7	94.1	97.3	85.2	81.8	88.7
SAM2Plus		95.7	94.1	97.4	85.5	81.9	89.1
SAM2.1		95.3	93.7	96.9	84.0	80.7	87.4
SAM2.1Long		95.7	94.0	97.3	85.3	81.9	88.8
SAM2.1Plus		95.8	94.1	97.4	85.6	82.0	89.3

**Table 3 sensors-25-04199-t003:** Performance comparison of 1080p videos in the LVOS v2 dataset based on A40 inference: SAM2, SAM2Long and SAM2Plus of all model sizes.

Method	Backbone	GPU Memory (MB)	Speed (FPS)
SAM2	Tiny	4165	30.87
SAM2Long		5707	13.99
SAM2Plus		4171	28.38
SAM2.1		4165	30.21
SAM2.1Long		5707	12.81
SAM2.1Plus		4173	28.16
SAM2	Small	4195	26.87
SAM2Long		5763	13.32
SAM2Plus		4205	26.63
SAM2.1		4193	26.65
SAM2.1Long		5811	12.81
SAM2.1Plus		4203	26.33
SAM2	Base+	4447	21.32
SAM2Long		5971	11.69
SAM2Plus		4453	20.35
SAM2.1		4449	20.11
SAM2.1Long		5971	11.71
SAM2.1Plus		4453	20.26
SAM2	Large	5497	13.69
SAM2Long		6779	8.47
SAM2Plus		5505	13.55
SAM2.1		5499	13.13
SAM2.1Long		6913	8.51
SAM2.1Plus		5503	13.46

**Table 4 sensors-25-04199-t004:** Cross-architecture benchmarking of LVOS validation split, employing subscript notations s/u to distinguish metric distributions between known and novel semantic taxonomies under standardized evaluation frameworks [41].

Method	LVOS v2 Val				
	J&F	$J_{s}$	$F_{s}$	$J_{u}$	$F_{u}$
LWL [42]	60.6	58.0	64.3	57.2	62.9
CFBI [16]	55.0	52.9	59.2	51.7	56.2
STCN [43]	60.6	57.2	64.0	57.5	63.8
RDE [44]	62.2	56.7	64.1	60.8	67.2
DeAOT [45]	63.9	61.5	69.0	58.4	66.6
XMem [46]	64.5	62.6	69.1	60.6	65.6
SAM 2 [2]	79.8	80.0	86.6	71.6	81.1
SAM 2.1 [2]	84.1	80.7	87.4	80.6	87.7
SAM2Long [4]	84.2	82.3	89.2	79.1	86.2
SAM2.1Long [4]	85.9	81.7	88.6	83.0	90.5
SAM2Plus	84.8	82.2	89.4	80.3	87.3
SAM2.1Plus	86.3	81.6	89.0	83.3	91.2

**Table 5 sensors-25-04199-t005:** Ablation on the effectiveness of the proposed modules. Performance is evaluated on 1080p videos from the LVOS v2 dataset using A40 GPU inference, with GPU memory usage and FPS reported as efficiency metrics.

Method	KF-IoU	AHFSS-DT	LVOS v2 Val			GPU Memory (MB)	Speed (FPS)
			J&F	J	F		
SAM2			80.2	76.8	83.6	4447	21.32
	√		81.2	77.5	84.8	4450	20.86
	√	√	82.9	79.3	86.5	4453	20.35
SAM2.1			83.1	79.6	86.5	4449	20.11
	√		84.5	80.8	88.2	4452	20.19
	√	√	86.3	82.6	90.0	4453	20.26

## Data Availability

All relevant data presented in the article are stored according to institutional requirements and, as such, are not available online. However, all data used in this manuscript can be made available upon request to the authors.

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
