# Peer review of "Improvement of SAM2 Algorithm Based on Kalman Filtering for Long-Term Video Object Segmentation"

_sensors, 2025, doi:10.3390/s25134199_

Round 1

Reviewer 1 Report

Comments and Suggestions for Authors

The paper titled "Improvement of SAM2 Algorithm Based on Kalman Filtering for Long-Term Video Object Segmentation" demonstrates a certain degree of innovation and practical value. However, in its current version, the manuscript still contains many aspects that require revision and supplementation. It is recommended that the authors conduct a thorough and comprehensive revision to further enhance the completeness and readability of the paper. The main issues are as follows:

1:The readability of Figure 1 needs improvement. Currently, the font size of the text in the figure is too small and difficult to read. In addition, there is overlapping text in the figure, which further affects its clarity. It is recommended that the authors increase the font size and rearrange the layout of the elements in the figure to ensure that all labels and annotations are clearly legible.

2:Section 2.3 "Kalman Filter" provides a review of various Kalman Filter (KF) variants; however, the authors ultimately adopt only the standard KF in their method. From the reader’s perspective, this section appears somewhat redundant and may divert attention from the main contributions of the paper. Unless the authors can clearly explain how this review of KF variants informs the method design, analysis, or experimental comparison, it is recommended that this section be removed.

3:Overall, the number of illustrations in the paper is relatively limited, and the existing figures are overly simplistic, making it difficult to effectively support the understanding of a complex model. It is recommended to remove the introduction of the SAM2 architecture in Section 3.1 “SAM2 Preliminaries” and instead shift the focus to a detailed presentation of the SAM2 Plus architecture, including a complete flowchart. Furthermore, to enhance the readability and persuasiveness of the paper, the authors are encouraged to add subfigures illustrating the encoder structure, the mask decoder, and the memory attention module. Alternatively, integrating the key components of SAM2 Plus into a single comprehensive architectural diagram would also help improve the overall clarity and professionalism of the paper, making it easier for readers to accurately understand the model design.

4:The first and last paragraphs of Section 3.3, “Adaptive Historical Frame Selection Strategy Based on Dynamic Threshold,” contain noticeable repetition, both excessively emphasizing the advantages brought by the proposed innovation. It is recommended that the authors merge and streamline these two paragraphs to avoid redundant expression.

5:The last two paragraphs of Section 4.2, “Qualitative Results,” contain clear repetition and redundancy. The authors repeatedly emphasize the innovativeness and superiority of their proposed method. It is recommended that these two paragraphs be merged and streamlined to avoid repetitive self-assessment.

6:It is recommended to adjust the order of Section 4.2 “Qualitative Results” and Section 4.3 “Quantitative Results.” In the current structure, the visual results are presented before the quantitative analysis, which appears logically inverted. It is advisable to first present the quantitative experimental analysis (i.e., move the current Section 4.3 forward) to demonstrate the effectiveness of the model through objective data, and then follow up with the qualitative results (i.e., move the current Section 4.2 after) to provide intuitive visual support. This structure aligns more closely with the conventional logic of academic writing and can enhance the overall persuasiveness and rigor of the paper.

7:In the current experimental section, the results for JF, J, and F metrics, memory usage, and FPS are scattered across different tables or paragraphs, lacking a unified presentation. It is recommended that the authors consolidate these key metrics—JF, J, F, memory consumption, and FPS—across all datasets into a single comprehensive comparison table. This format will help readers clearly grasp the balance and advantages of the proposed model in terms of performance, efficiency, and resource consumption.

8:In Section 4.3 “Quantitative Results,” it is recommended that the authors first clearly specify the code execution environment and computational hardware configuration used in the experiments, including key details such as the operating system, GPU model, and Python version, to facilitate reproducibility by other researchers. In addition, for the models being compared, the authors are advised to maintain and explicitly state the consistency of key hyperparameters, and to provide detailed parameter settings in the paper. This will not only ensure the fairness of the comparative experiments but also significantly enhance the credibility and persuasiveness of the results.

9:From Table 2 and the description below, the authors indicate that SAM2 Plus slightly increases video memory usage compared to SAM2 while maintaining the same inference speed, concluding that SAM2 Plus is more suitable for real-time applications. However, this statement contains a logical inconsistency: an increase in memory usage generally affects real-time performance to some extent, especially in scenarios with limited hardware resources.

10:The manuscript repeatedly cites the same references multiple times. It is recommended that after adequately citing relevant literature in the earlier sections, the authors avoid unnecessary repeated citations later in the paper.

11:In Section 4.4 “Long Video Segmentation in Real Scenes with SAM2Plus,” the authors apply SAM2Plus to experiments in real-world scenarios. It is recommended to provide a detailed description of the hardware configuration used for the experiments as well as the specific experimental procedures and settings, rather than merely presenting individual detection results. This will help readers better understand the actual experimental environment and operational details, thereby enhancing the credibility and persuasiveness of the results.

12:The paper emphasizes in the results analysis section that the method achieves a good balance between accuracy and real-time performance. Based on this, it is recommended that Section 5 “Discussion” not only presents the impact of each module on accuracy but also includes evaluations of inference speed and memory usage, to comprehensively reflect each module’s contribution to model efficiency. Furthermore, given that the authors frequently compare their method with SAM2Long, it is advised to include ablation experiments that progressively integrate each module into the SAM2 baseline and directly compare their performance with SAM2Long.

Author Response

Our specific responses to the revision suggestions of the thesis are attached.

Reviewer 2 Report

Comments and Suggestions for Authors

This work concentrates on developing Visual object tracking (VOT) using Segment Anything Model 2 (SAM 2), a latest model. Specifically, the proposed method addresses limitations of long-term object tracking by utilizing adaptive historical frame selection strategy based on dynamic threshold, and memory management and optimization for efficient video segmentation. While the paper presents reasonable approaches validated by experimental results, the current manuscript does not effectively convey the significance of the proposed method in VOT, particularly regarding the technical aspects and contributions related to memory efficiency and long-term tracking challenges. My comments are as follows:

- Although Sec. 2.3 (and Sec. 1) covers the Kalman filter and some related works, the authors are suggested to provide a brief overview of related works that address similar challenges (e.g., occlusion or re-identification) in the broader context of MOT, such as Multiple Hypothesis Tracking (MHT) [R1], Joint Probabilistic Data Association (JPDA) [R2], and Labeled Random Finite Set (LRFS) [R3], in between the Kalman filter and visual MOT. For example, it is observed that state-of-the-art LRFS methods show robust performance in fields related to computer vision. 

- In Sec. 3.1, a succinct analysis of time complexity is provided. This analysis should be elaborated to provide a better understanding of the proposed method. Further, its placement in this section seems questionable. It might be more logical to present a more detailed complexity analysis after describing the full algorithm.

- Sec 3.4  appears to be overly subdivided into numerous (sub)subsections. For improved readability and paper structure, consider merging some of these subsections. Additionally, the simplified figures (Figs 4 and 5) should be enhanced for clarity and illustration of the proposed techniques.

- More detailed justifications (rationales) for each specific component utilized in this work are required. 

- It would be beneficial if all terms/symbols were clearly defined or clarified. Moreover, if the work does not include information about source codes, pseudo codes are strongly recommended in the manuscript. This would enhance reproducibility, which allows potential readers to better understand and replicate the proposed methods and experimental comparisons. 

- As a final minor comment, careful formatting is required as there are inconsistencies throughout the manuscript. For instance, unnecessary repeated sentences and duplicated abbreviations make the paper less professional. 

[R1] S. S. Blackman and R. Popoli, Design and Analysis of Modern Tracking Systems. Norwood, MA, USA: Artech House, 1999.
[R2] Y. Bar-Shalom and T. E. Fortmann, Tracking and Data Association. Orlando, USA: Academic Press, 1988.
[R3] R. Mahler, Advances in Statistical Multisource-Multitarget Information Fusion. Artech House, 2014

Comments on the Quality of English Language

Please see my suggestions.

Author Response

Our specific responses to the revision suggestions of the thesis are attached

Round 2

Reviewer 1 Report

Comments and Suggestions for Authors

The overall structure of the paper is clear, the research content is somewhat innovative, the methodology used is reasonable, and no obvious problems have been found, so I recommend that it be accepted. It should be noted that: while reviewing the content, I have also carried out a preliminary check on the English grammar and diction in the text. However, it is recommended that the authors still ask a professional language editor to touch up the whole text before submitting the final manuscript, so as to ensure that the language expression is more accurate, standardized and fluent.

Comments on the Quality of English Language

The overall structure of the paper is clear, the research content is somewhat innovative, the methodology used is reasonable, and no obvious problems have been found, so I recommend that it be accepted. It should be noted that: while reviewing the content, I have also carried out a preliminary check on the English grammar and diction in the text. However, it is recommended that the authors still ask a professional language editor to touch up the whole text before submitting the final manuscript, so as to ensure that the language expression is more accurate, standardized and fluent.